# Family's perceptions of their members who use nyaope in Tshwane, South Africa

**Doudou K. Nzaumvila** [1,2*], **Robert Mash**[1], **Toby Helliwell**[3]

**1** Department of Family and Emergency Medicine, Faculty of Medicine and Health Sciences, Stellenbosch University Cape Town, South Africa, **2** Family Medicine and Primary Health care, Sefako Makgotho Health Sciences, Pretoria, **3** General Practice Department, School of medicine, Keele University, London, United Kingdom

* doug_nk@hotmail.com

## Abstract

### Introduction

Over the last two decades, nyaope use has evolved to become a prominent substance use disorder in South Africa, posing a significant public health burden. The majority of users are young people who are solely concerned with their next nyaope joint. This study aimed to explore the perception of family members on the factors associated with the use of and dependency on nyaope.

### Methods

This was a descriptive exploratory qualitative study conducted in Tshwane, South Africa. Data were collected from 32 family members of Nyaope users via three focus group interviews conducted by a retired psychologist nurse in the three townships of Tshwane.

### Results

The findings revealed a complex and interconnected web of elements that shape the journey of individuals from the onset of nyaope use to the point of dependence and eventual departure from their family homes. Rather than following a linear path of events, this pathway is characterised by a dynamic interplay of seven distinct themes, namely concealed nyaope use, family concerns and suspicions regarding nyaope use, confirmation of nyaope use, possible reasons for using nyaope, barriers to obtaining assistance for nyaope users, family distress, and the transition from home to a life on the streets.

### Conclusion

Most users ended up being disconnected from their families. Family members' opinions noted that the problem is perceived to be a web of elements working together rather than a linear path of events. The findings have implications for substance use services, social services, health and police services as well as schools.

**Data availability statement:** The restriction to data is enforced both directly by the participant

consent and indirectly by the Stellenbosch University Human Ethic Committee, which authorized us to conduct the study. Due to the possibility of retaliation from drug (nyaope) dealers in the communities where we collected data, the consent form clearly protected the participant's identity, and participants agreed to participate in the study knowing that the data collected would be protected by a password known only to the researchers, used only for research purposes, and would not be made public. Furthermore, the collected data contained potentially sensitive information and highlighted several legal gaps, exposing officials. Much like the participants' perception that certain police personnel collaborated with nyaope suppliers. For data access, please contact Charmaine Khumalo, Head of Health Research Ethics, at 021 938 9075; ckhumalo@sun.ac.za. Or contact Mrs. Brightness Nxumalo, Coordinator, Health Research Ethics Committee 2 (HREC 2), at 021 938 9207; brightness@sun.ac.za.

**Funding:** The author(s) received no specific funding for this work.

**Competing interests:** The authors have declared that no competing interests exist.

## Introduction

Over the last two decades, nyaope has emerged as a major substance use disorder (SUD) in South Africa (SA) [1–4], thereby posing a substantial and ongoing public health concern [1,5]. This is in a country already grappling with many other substance use disorders [6,7]. nyaope has become a common street substance in underprivileged townships, such as in the Tshwane district [4–8].

Nyaope, a recreational street substance that is uniquely South African, is primarily made of low-grade heroin, and seen as a new psychoactive substance [9]. The amount of low-grade heroin in each batch varies greatly, because it is not manufactured according to a fixed recipe [10,11]. It is also mixed with other substances, such as milk powder, rat poison, bicarbonate of soda, cannabis or antiviral drugs [10,11,12].

The use of and dependency on nyaope, which is highly addictive, has severely impacted individuals, families and communities in Tshwane [1,4,5,9,13]. Users may drop out of school and higher education, or end up retrenched from work. They may find themselves on the street, hustling for cash to buy nyaope, often in illegal ways [4]. Most of the users are young people who focus only on their next nyaope joint [4,9,13,14].

Nyaope affects people at all income levels, but it has the greatest effects on low-income families who are constantly dealing with other psychosocial problems [4]. Conflict between family members, financial pressure as well as emotional and psychological stress have been documented [1,15–17]. One of the outcomes is that users lie to and steal from their relatives to obtain cash to service their habit. The detrimental impact of substance use can also affect the family's neighbours and surrounding community [18].

There is a small but growing body of literature on nyaope use and the impact on families [5,19,20]. Parents of nyaope users as well as the users themselves could benefit from therapeutic interventions, because they are all psychologically affected [19]. It is documented that family members and close friends of nyaope users are more likely to experience varying degrees of depressive symptoms [5]. Householders rarely seek medical assistance and are not screened for depressive disorders, according to the most recent research, therefore, not diagnosed or treated, which lowers their quality of life [5].

Rehabilitation facilities have limited capacity, and it has also been reported that less than 3% of nyaope users who finish therapy experience a full recovery [21]. Over 40% of them leave the programme before completion [21]. The available evidence does not seem sufficient to enable long-term solutions, leaving most families feeling hopeless [19, 20]. It has become essential to comprehend the viewpoints of family members of nyaope users in an attempt to identify key factors that lead to dependency and develop better interventions, such as opioid substitution programs [22].

Although there are many factors identified as contributing to SUDs in general [23] and there is a growing body of research in the field [22–24], there is still a lack of evidence on nyaope specifically [20]. It goes without saying that prevention is preferable to treatment because it frees up more resources for those who actually need them. Therefore, it is important to consider the various risk factors and drivers of initial nyaope use. This study is a component of a larger investigation that aims to identify factors that influence nyaope use in Tshwane. Unfavourable home environments, mistrust between community members and the local police, easy access to nyaope at school, inadequate social services, a lack of religious or spiritual inclination, unfavourable community environments and the effects of nyaope on users were some of the factors from a previous study focusing on the perspectives of community member [1]. This study aimed to explore the perspectives and experiences of family members of Nyaope users on the factors underlying the use of and dependency to nyaope in Tshwane.

## Methods

### Study design

This was a descriptive exploratory qualitative study that collected data by means of semi-structured focus group interviews (FGIs) with family members of nyaope users in Tshwane, South Africa.

### Setting

This research was carried out in Tshwane, a multiethnic metropolis in Gauteng Province, South Africa, which is home to an estimated 2.3 million people, most of whom reside in townships [25]. Like many other cities, Tshwane is plagued by socioeconomic problems such as poverty and income inequality. Certain neighbourhoods have problems, including informal settlements, inadequate housing and a lack of basic services, while other neighbourhoods have well-developed infrastructure and acceptable housing. A critical issue is housing affordability, especially for low-income households [25].

There is a community-based substance use programme (COSUP), offered by a non-governmental organisation, which attempts to address the nyaope problem. This is an outpatient programme that emphasises harm reduction strategies and clinical care for nyaope users rather than prohibition and abstinence. It offers physical, mental and substance use screening, assessment, brief interventions and harm reduction counselling, opioid substitution therapy, needle and syringe exchange, social services, skills development and shelter. The above services are currently not available for SUD in public health services [21].

### Study population and selection of participants

The study population was defined as family members of nyaope users (>18 years age), who had lived in the same house as the user for at least a year, did not use nyaope themselves, and lived in Tshwane district. They were excluded if they had participated in the previous study [1].

### Sample size and sampling

At the 17 COSUP sites, users were accompanied by their family members. Social workers took the opportunity to purposefully identify potential participants and extended invitations for them to join the study. Tshwane is home to a diverse population that speaks a variety of languages (Afrikaans, English, Si Tswana, Si Pedi, Si Zulu). Participants were free to utilize the language of their choice, as the interviewer was fluent in all of them. There was no linguistic discrimination present during the selecting procedure. Participants were specifically chosen due to their enthusiasm to share their opinions on the matter. Calls were made to prospective participants to remind them to attend the FGI. The authors planned 2 to 3 FGIs with 9 to 12 participants, with the final sample size determined by saturation of data.

### Data collection

Based on the risk and protective variables for substance use included in the National Drug Master Plan 2013–2017 [26], the Bronfenbrenner's ecological theory of development [27], and in keeping with the study's objectives, the authors developed an interview guide. Each participant completed a written consent form after being informed of the study's objectives. The opening question was: "Tell us as much as you can about the risk and protective factors you have observed surrounding the use of Nyaope for your relative." The intended topics listed in the guide were on individual, family, community and school risk and protective factors.

Participants were invited to discuss any relevant nyaope-related topics. By utilising open-ended questions that allowed participants to elaborate on subjects, they were given the chance to share their opinions. A nurse and clinical psychologist, with training in qualitative interviewing and fluent in local languages (Afrikaans, English, isi-Tswana), led the tape-recorded FGIs from March the 1st, 2021 to May the 31st, 2021. Three Tshwane townships – Eersterust (at the Eerster-ust Community Health Centre), Nelmapuis (at a participant's home) and Shoshanguve (at a local church) – were the sites of the FGIs. The interviews lasted for approximatively 90 minutes.

## Data analysis

The authors used the service of a professional linguist who translated vernacular audios and transcribing straight ahead to English. To analyse the data, the framework method was used in a step-by-step fashion [28,29] with the assistance of Atlas-ti software [30].

The researcher began by becoming acquainted with the data and field notes. Following that, the researcher inductively created a coding index, which was then organised into categories. The researcher then coded all the transcripts. Next, the researcher generated reports based on the categories to bring all the data together. Finally, the data were interpreted to identify themes. The breadth of the perspectives and experiences was interpreted as well as any connections between themes.

## Trustworthiness

The first author (DN) coordinated and led the research, which was supervised and peer-reviewed by the other authors (RM and TH). They particularly focused on the construction of the coding index and interpretation of the data.

Although the person who performed the FGIs lived in one of the townships affected by nyaope, she possessed sufficient interviewing skills to neither alter nor influence the participants' responses. Furthermore, the main researcher (DN), an academic family physician, had previously dealt with nyaope users in various Tshwane COSUP centres, communicating with users in English. He was involved in the medical assessments of users before OST initiation at some COSUP sites, ensuring the safety and appropriateness of OST. As an experienced family physician with a background in qualitative research, he added value to this research process. He was well aware that positive bias could occur during data processing. He was conscious of this during the data processing procedure.

## Ethical considerations

The study was authorised by the Health Research Ethics Committee at Stellenbosch University, with reference number S20/04/092. COSUP management also granted permission to conduct this study. Because the subject matter is sensitive, conducting such an interview may elicit strong emotions. We mentioned to the participants that they might withdraw at any point of the interview or even after the interview if they felt like it, and that the social workers at COSUP were available for debriefing, if necessary, after the interview.

## Results

The 32 participants represented a variety of age groups, genders and relationships to the users, as shown in Table 1. However, the majority were older women, who were mothers or grand-mothers of the nyaope users. Saturation of data was thought to be achieved in the third FGI, as no new themes were emerging from the data.

The thematic analysis identified seven key themes that align with a pathway from initial nyaope use to living on the streets (Table 2).

**Table 1. Characteristics of participants.**

| Characteristics of participants | FGI 1 | FGI 2 | FGI 3 | Total |
|---|---|---|---|---|
| **Gender** | | | | |
| Female | 9 | 11 | 9 | 29 |
| Male | 2 | 1 | 0 | 3 |
| **Age groups (years)** | | | | |
| 20–29 | 0 | 1 | 1 | 1 |
| 30–39 | 2 | 2 | 0 | 6 |
| 40–49 | 5 | 5 | 4 | 14 |
| ≥50 | 4 | 3 | 4 | 11 |
| **Categories of participants** | | | | |
| Aunty | 1 | 0 | 0 | 1 |
| Cousin | 0 | 1 | 0 | 1 |
| Father | 1 | 1 | 0 | 2 |
| Grandmother | 3 | 6 | 3 | 12 |
| Mother | 5 | 4 | 5 | 14 |
| Niece | 0 | 0 | 1 | 1 |
| Sister | 1 | 0 | 0 | 1 |
| **Total** | 11 | 12 | 9 | 32 |

**Table 2. Themes.**

| | Themes |
|---|---|
| 1 | Concealed nyaope use |
| 2 | Family concerns and suspicions regarding nyaope use |
| 3 | Confirmation of nyaope use |
| 4 | Possible reasons for using nyaope |
| 5 | Perceived barriers to assistance for nyaope users |
| 6 | Family in distress |
| 7 | From home to living in the street |

## Theme 1: Concealed nyaope use

Families reported that there were few predictive or early warning signs and that users were often secretive, making it difficult to detect the initial signs and symptoms of nyaope use. The use of nyaope came as a surprise, especially if users were previously seen as *"good children"*. The sudden revelation of a family member's involvement with nyaope was described as shocking, because it challenged the perception and image they had previously held. A mother of a user, while shedding a tear, had this to say:

> *"… He used to be a very disciplined child, respectful, does his chores and he would never backchat …" (Mother 3, FGI 1).*

Some participants explained that they were shocked, confused and had a sense of betrayal, as they did not see it coming. nyaope users employed various tactics to hide their new habit and maintain a façade of normality, particularly with their parents. An aunty of two nyaope users described how they hid their initial involvement with nyaope:

*"… Before nyaope, they were there, they were our children, brothers and cousins to our children … I would say that people who are using nyaope are dissimulators, very clever. I think nyaope was designed for smart people … I usually tell people that nyaope was not designed for dumb people or slow people, it was designed for smart people …"* (Aunty, FGI 1).

## Theme 2: Family concerns and suspicions regarding nyaope use

Family concerns and suspicions that there was something wrong were related to new aggressive behaviour, stealing, decline in personal hygiene, self-isolation, truancy and a significant drop in school performance. Aggressive and angry behaviour might scare family members and be associated with denial about nyaope use:

*"… He was aggressive, he was a different person. We were even scared of him now, thinking that if you say something to him he would do something terrible …"* (Mother 5, FGI 3).

In the early stages of nyaope use, participants reported minor pilfering of small, easily concealed items, which could easily be sold to obtain quick cash. Later on, this escalated to stealing more valuable items, as the financial needs increased. These stolen items were sold at significantly reduced prices to quickly generate cash:

*"He sold the microwave. He has twins who are six years. On New Year's evening, his father and the twins asked me where the fridge is. I asked them, fridge? When I looked, the fridge was not there. He went and sold the fridge for R300. I called another police officer. When his father went to look for him, he found him and he told him to take him where the fridge was. He had sold a double-door fridge for R300..."* (Mother 2, FDI 2).

Some family members noticed the withdrawal and disconnection of nyaope users from family life, and that they spent more time outside the family unit. One mother remembered this:

*"… He will eat once, for the whole day. He used to spend time with us and now we are annoying him … I am annoying him because there is something he needs out there and he can't even tell me …"* (Mother 1, FGI 3).

Most of the participants noted the neglect of personal hygiene that started with a bad body odour:

*"… The other thing I saw in the house when he started was the smell, a unique smell/odour. Even if he can take a bath 3 or 4 times, he will still have that unpleasant smell …"* (Father, FG2).

Family members became suspicious of the use of nyaope when schools reported truancy and a decline in school performance. Users might also attempt to hide the decline by faking their school reports:

*"… Then, one day, they called me at school and said your child had created his own school report. Because when the teachers mark his schoolwork, he got 20% and so, but in his report it was As and Bs … He and his friend found someone to make reports for them …"* (Mother 2, FGI2).

### Theme 3: Confirmation of nyaope use

Some participants felt it was crucial to corroborate their strong suspicions that their child was using nyaope. Some used confrontation, others tried to gather tangible evidence, but most of the time the users denied using nyaope. One parent explained that denial might be related to the stigma associated with nyaope use in communities:

> "… *Because now it is nyaope and to us as the society when you associate with people who use nyaope, it is like an abomination …*" (Father, FGI 1).

Many continued to deny use of nyaope even when directly confronted by family members. Some participants clearly voiced how difficult and distressing it was to confirm that a relative was using nyaope so that they could help them:

> "… *Then I asked him, how can I help you, my child, if you still denying it? He said why do you want to help me because I don't smoke nyaope …*" (Grandmother 1, FGI 3).

One mother described how she went through her daughter's belongings to gather evidence of nyoape use:

> "… *First, it was the cigarette, then the second one I think was weed, patje, dagga (marijuana). Then the third one was nyaope … I did find a plastic one, the powder one … in her bag. I always search her bags when she is around at home … I also found a box of match sticks and weed …*" (Mother 1, FGI3).

Although denial and refusing assistance was the initial experience of family members, eventually some users might reach a point where they admitted use and asked for help:

> "… *I was so hurt; I was so disappointed and emotional. I was hurt, until 2014 when she went to rehab … She took the first step and said she wants to go to the clinic, she wants to start afresh …*"

### Theme 4: Possible reasons for using nyaope

Possible reasons for using nyaope were categorised as factors external to the family, internal to the family, or personal factors related to effects of the drug itself. Participants were very vocal on external factors such as peer pressure, the school environment and intimate partner pressure. One mother cited her son's friends:

> "… *According to my experience, what causes him to start using nyaope, it could be friends … I think the peer pressure from friends contributed a lot. He lived a lifestyle where he had friends whom he used to go out with a lot …*" (Mother 3, FGI 1).

Many participants expressed their resentment that the school environment was not safe for children:

> "*That is why I am saying school is a dangerous place for the kids... So, at school, some of the behaviours are out of line and they pass on to our kids. I am not saying that education is dangerous; I am saying that the school environment is where they get into wrong things …*" (Mother 1, FGI1).

Some family members of female nyaope users reported that their boyfriends influenced them. A cousin of a female nyaope shared this:

*"… According to my experience, what causes one to start using nyaope could be friends. The people they spend time with, for example, boyfriend and the friends they spend time with. However, according to me, a boyfriend influenced my family member. He was putting her under pressure, saying because I smoke you should also smoke …"* (Mother 2, FGI3).

She elaborated further, mentioning the possibility of forced prostitution by the partner as a way of financing the habit:

*"The reason their boyfriend pressurises her into smoking is because they are always hustling. So, he will say to the girl that they can't hustle alone … because you are a female and you have a body you also need to hustle … I don't get anything; you must also make a plan … because she wants to please the boyfriend by selling her body… the boy doesn't care as long as the girlfriend brings money and they can buy another fix …"* (Mother 2, FGI13).

Traumatic events within the family might lead to nyaope use as a coping strategy:

*"From my side, it is my cousin who is using. I think the reason he ended up using is because of the treatment he received from home..."* (Cousin 2, FGI2).

This was also the case with another woman after the passing of her husband:

*"… My son started using drugs after my husband passed away …"* (Mother 3, FGI3).

Some participants reported that users enjoyed nyaope as it gave them a positive feeling. This mother shared her son's experience:

*"… He started enjoying being high and it happened for a few days and learning that when I smoke this, I don't get withdrawal symptoms … after he smoked now, he is aware that the drug (nyaope) makes him feel good again and again"* (Mother 3, FGI3).

Other participants revealed that their relatives used another substance that made them very active; they then used nyaope as a downer and became dependent:

*"… Yes, the one he has at that time, which is rock or crystal meth, which makes him hyperactive. So, he uses nyaope as a downer to make him sleep …"* (Mother 2, FGI 1).

## Theme 5: Perceived barriers to assistance for nyaope users

The participants revealed that they were in a desperate, but futile search, for long-lasting solutions to nyaope use. Family members vehemently described the mistrustful relationships they had with the police. They voiced their dissatisfaction with their local police stations. The fear of retaliation and reprisal appeared to be a significant barrier to reporting nyaope businesses and activities in the communities. The belief that the police may be cooperating with the nyaope sellers further reinforced this fear and could discourage parents from taking action. One parent had this to disclose:

*"Police are aware that these people are selling Nyaope; they are involved too … And we know them, it is just we are scared to talk... Because when you report them, the police will tell them who reported them …"* (Father, FGI 2).

Parents might be discouraged from seeking police assistance because they perceived the police as being passive when it came to dealing with the issue of nyaope. While some people had negative encounters with the police in relation to drug-related problems, it did not necessarily mean that all police officers were passive or unresponsive:

*"What pains me is that this nyaope is sold in the community. We know where they are selling it but, at the end of the day, when we report to the police or wherever, there is nothing done about it to help our community and the children who are using nyaope …"* (Father, FGI 2).

Dissatisfaction with the police response to nyaope and other drug-related problems led to a lack of trust and discouraged them from seeking assistance:

"*I once took my son to the police station. I wanted them to arrest him. The station commander said to me that I am just wasting my time to bring my child here so that they can arrest him.*" (Mother 4, FGI 2).

She went on to say that she had been informed that the issue does not simply involve the police, but also the judicial system:

"*He said the magistrate releases these children at court and yet you are bringing yours here to be arrested for something for which he is not the first and certainly not the last. The magistrate releases the children from court and tells them to go, telling us as the police to take them back where we found them, so they are making it hard for us to arrest them. So that is why we are no longer arresting children who are using nyaope; we are no longer arresting people who are selling nyaope …" (Mother 4, FGI 2).*

As a result, users were not concerned about the potential repercussions of their illegal activities and use of nyaope, even though it is now criminalised. This is what one cousin of a nyaope user shared:

"*He will say I am not scared of the police; you can get me arrested. He will take the table and throw it there and it will break*" (Cousin, FGI 2).

Another participant concurred and said

*"Mine will say the police will take him now and he will be out in no time. He doesn't sleep in the holding cells"* (Mother 1, FGI2).

She outlined the possible reasons why the police are unable to hold them for an extended period:

*"No, when they (nyaope users) get to the holding cells, they scream and in the evening they will release them. Because they have stomach cramps. They will start screaming because they want to smoke. This is what happens every time when they get arrested"* (Mother 1, FGI2).

Parents voiced their dissatisfaction with the Tshwane-based rehabilitative services. They felt hopeless and were discouraged by the poor results of rehabilitation:

*"To tell the truth, I never took mine to rehab. Because, while I was in a process of taking him to rehab, I saw many who are from rehab going back to smoking. So that discouraged me … I think they discharge them before the time. If there could be strict measures because this thing of six weeks … It is too short for a person to detox and get rid of all of that …"* (Father 1, FGI 1).

He emphasised the lack of ongoing support after the rehabilitation programme, as users return to the same milieu of peer pressure and unemployment, and relapse repeatedly:

*"So, our children, when they come back from there, they go and mix with those who are still smoking … Yes, they are their friends but … if they could get job when they come back... If they go to work or wherever, it was something better. But now when they come back, they mix up with them again."* (Father 1, FGI1).

## Theme 6: Family in distress

Some participants reported that dealing with a loved one who is using nyaope is extremely challenging and emotionally draining for parents. The impact on the family dynamic seemed overwhelming, and it was difficult to continue to provide the necessary support while also taking care of one's own well-being. One parent shared this:

*"I think I will beg to differ a little from what you guys said because what I have experienced with my son is that we gave him so much love in the family. We treated him like any other child at home. We never showed him that we can see his mistakes. It took time before we told him that he is messing things up in the house … We couldn't tell him straight. Things went missing from home, but we couldn't tell him that it was him … We tried to give him love … So, it came to a point where I felt like, I am a human being, yes, I feel for you, but if you don't feel for me, this makes me angry."* (Father, FGI 3)

Respondents were united in describing their distress and suffering, brought on by the use of nyaope. Families became dysfunctional; this might lead to consequences such as conflict with neighbours or the users' being rejected. Some parents chose to purchase nyaope for their children in order to prevent conflict within the family or with neighbours due to stealing:

*"I am one of those mothers that gives him money – not to buy drugs but to leave people's things alone. I don't have time for police cases and all that. I can't stay off my work for all this nonsense …"* (Grandmother 3, FGI3).

Conflict with neighbours was common; families felt stigmatised, and users were sometimes exploited:

*"When it comes to neighbours and you have a child who is using nyaope, you are a laughing-stock …"* (Mother 3, FGI 1).

*"They will call him and send him to go and buy cold drink, and then later on they will send him to go and buy milk. Why can't they send him once with a list of all the things they need? But, they will keep on sending him up and down on the hot sunny day just for them to give him R2. Furthermore, whenever a neighbour loses something, the child is the first to*

*be accused of stealing … So every time they will want to blame my child and I end up being angry … not talking to them … they point everything to my child and I ended up in a conflict with them... They say you don't discipline him enough. I ended up taking wrong decisions from the pressure from the neighbours …" (Mother 3, FGI 1).*

Family dysfunction, distress and conflict with neighbours could lead to the user being rejected or ejected from the family home. This might also depend on the family's beliefs and values as well as their experience of the user:

*"… I told her I don't want to get involved; she must go alone and do whatever. I told her that I want to see and not hear that she has changed … I didn't want to see her anymore …" (Mother 3, FGI3).*

### Theme 7: From home to living in the street

Eventually the family's distress, loss of compassion and rejection of the user led the user to homelessness and living on the street. Users were isolated, desperate and detached from their family and social bonds. However, some parents perceived that nyaope users found a second home in the street:

*"We shouldn't make them choose between the streets and the family. We must recognise them. Each of those children, when you ask them, they have a story from home and their stories are similar. You will ask them if they want to quit and one of them will say yes but my mother doesn't love me. I want to quit but my uncle sold our house."*

The *"nyaope brotherhood"* refers to a group of individuals who are using nyaope and living on the streets. Some participants expressed their concerns and perceptions regarding the implications of their relatives joining this "brotherhood". This is what a mother who successfully returned her child to her home had to say:

*"I personally think as the family of these people we shouldn't forget that we are in the battle with the streets. The streets are showing love to our kids or our cousins or our brothers. We must show love to them. We must understand that they are alone in the streets and the friends encourage them on wrong things. So, we must show them love so that when they are with family they feel that there is no difference with the streets, the only difference is that there is no nyaope at home …" (Mother 1, FG1).*

As users lived on the streets and became more connected with each other, they became more and more disconnected from the broader community. Community members were concerned for their property and personal safety, and fear of nyaope users led to their victimisation. Parents saw this vicious circle leading to victimisation as another barrier to assistance:

*"So, if the community can stop being cruel to them and stop calling them all sorts of names, recognise them and give them jobs and pay them money which is worth their lives, and stop giving them R20, they will see that we care enough. They want to quit but we are the problem. We are oppressing them and we don't want them near us" (Mother 2, FGI 3).*

### Discussion

This study highlighted the pathway from initial use within the family to be being absorbed by the nyaope brotherhood and or living on the streets. Fig 1 provides an overview of the

study's main results and emphasises the identified possible central role of family, along with other stakeholders, as seen before [1]. Various perspectives are included this time, including negative home environments [1,4,5], mistrust between the local police and the community [1], easy access to nyaope at school [1], insufficient social services [1,4,5,8,13,22], detrimental community environments, and nyaope's effects on users [1].

## Discussion of findings

The discussion plan is organised around the seven themes, which support the conceptual framework shown in Fig 1. The thematic analysis presented an initial picture of several obstacles, that prevent help from being accepted and allows dependence on nyaope to develop. At each point along the subsequent pathway to living on the streets, one should question why nothing was done. The term nyaope brotherhood was coined by the authors to characterise the nyaope user's subculture as a distinct group from the general public, living on the streets with its own set of customs, beliefs and behaviours. The authors unveiled the inherent barriers to addressing nyaope use dependency: four of the seven themes are directly or indirectly obstacles to any possible assistance (concealed nyaope use, confirmation of nyaope use, perceived barriers to assistance for nyaope users, from home to living in the street).

The representatives of the seven themes' daily activities were grouped together. Home (red) speaks for the home setting, where peer pressure, the educational setting, and coping strategies are some of the factors that lead to the first use of nyaope. Families may respond by confronting the user, rejecting them, or denying them. As a convergent point that functions as a black

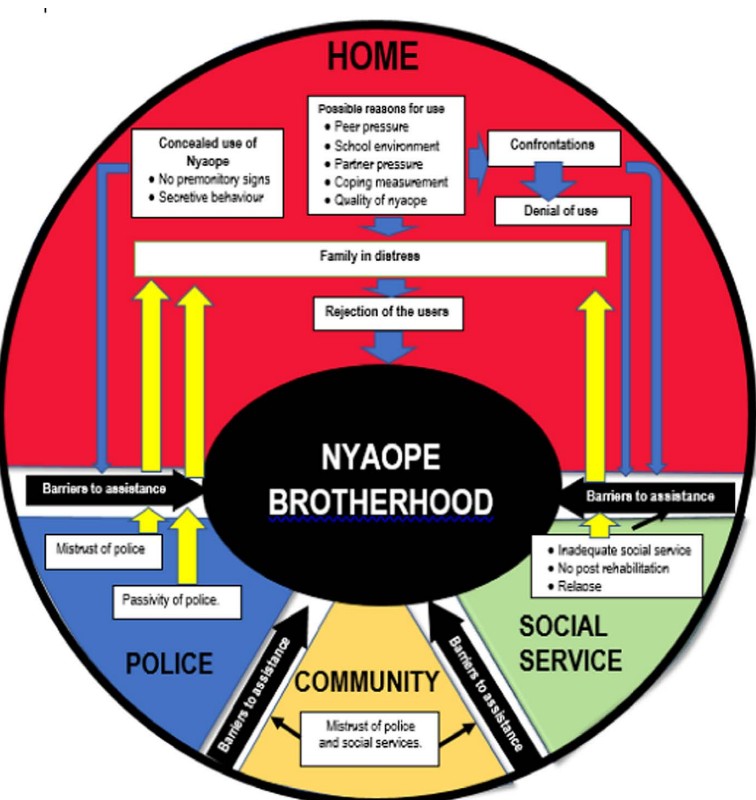

**Fig 1. Emerging concept of elements of elements driving development of nyaope addiction.**

hole where all users terminate, the nyaope brotherhood (black) frequently reinforces nyaope usage and makes it challenging to overcome dependency. Police (blue) shows the function of law enforcement, emphasizing obstacles to help including police passivity and mistrust. Community (yellow) shows the larger social context, highlighting the mistrust of social services and the police in the community. Social service (green) indicates social support networks and rehabilitation, highlighting obstacles such as early discharge, insufficient services, and a lack of post-rehabilitation programs. Black arrows stand for obstacles that prevent effective intervention and rehabilitation, treatment and recovery. Blue arrows display family-user encounters that result in confrontation, denial, and rejection, which either directly or indirectly foster the Nyaope fraternity through obstacles for assistance. Yellow arrows indicate routes of building up family distress as they cannot receive aid from the police or social agencies.

One of the biggest barriers was the concealed use. We found that at an early stage, it is used secretly. Most of the relatives became aware of the problem only once it was too late as the users were already showing signs of dependency on nyaope. This is in keeping with literature [20]. Although secrecy is known as a characteristic of SUD [31], the intense negative social impact of nyaope use [1,4] may contribute to the dissimulation attitude. At a later stage, when physical signs of nyaope use and dependency are visible, users may isolate themselves to hide physical signs and may regulate their behaviour to avoid suspicion. Our findings concurred with previous studies that nyaope users often isolate themselves from their parents, spending significant amounts of time away from home or in the company of peers involved in nyaope use [1,4,13]. Authors consider the use of concealed nyaope and self-isolation as an indication of the user's continued emotional ties to his family as well as respect for parental and society norms and authority. In reverse, this creates a barrier to assistance.

Another barrier to assistance is the denial of use, which is not new in the context of SUD [31]. Our findings suggested that families were concerned and suspicious about nyaope use. Unlike some literature that suggests that users may cry out for help [4,13,22,23], our study highlighted parents' report of users often denying their habit even when confronted with overwhelming evidence and are unlikely to reach out for assistance. Delay in identifying nyaope use may pose significant challenges in obtaining help and delaying the path to recovery.

Untrustworthy interaction between police and parents is not new in nyaope literature [1,20]. Users must get their fix on a regular basis [2,4,13]. To get cash, users resort to stealing, without fear of the police, as they are not arrested, or released without charge. This could be done to allow them to avoid withdrawal syndromes whilst in custody, creating a perception that police are passive. This creates a barrier to assistance that could be solved by a collaboration between the police and medical services to provide assistance to individuals experiencing withdrawal symptoms while in custody. This will resonate with recent publication [32]. This collaboration should be added to already existing initiatives, such as the exchange of needles and syringes or the opioid treatment [14,21–23], which may appear suspect to the police.

Previous studies [20], along with our findings, suggest that parents may experience a sense of relief when nyaope users leave home. Participants reported that nyaope users might have sought out alternative social circles that align with their dependence of nyaope use: the nyaope brotherhood. These new social connections often revolve around fellow users and individuals who facilitate access to nyaope [13,20,23]. The authors argue that all the inherent barriers mentioned above could facilitate the formation of the nyaope brotherhood. The all-encompassing nature of the brotherhood may foster a sense of belonging within the group while distancing individuals from their previous social and family connections.

### Strengths and limitations of the study

The unveiling of the inherent barriers to assistance that feeds the nyaope brotherhood is a valuable insight to the existing body of knowledge on the use and dependence on nyaope. Purposeful sampling ensured a broader inclusion of typical South African household individuals who are affected by the phenomenon, with different relationships to the users. This study had certain limitations that should be considered when interpreting its findings. Firstly, the research focused on a specific subset of households, primarily from a particular region and socio-economic background. While this approach allowed for in-depth exploration, it may not capture the full diversity of experiences and perspectives among nyaope users and their families across different contexts. The transferability of the above should be recognised as a limitation. Secondly, the stigma and illegal nature associated with nyaope use could have influenced participants' responses. The fear of judgment or legal repercussions may have led some participants to provide more socially acceptable responses or underreport information related to nyaope use. Thirdly, it is also worth noting that the study participants were mainly women (mothers and grandmothers), which may have limited the depth and breadth of data saturation, as other family members' perspectives and experiences were not as prominently featured.

### Implications of the findings

Overall, the implications of a study on the perceptions of family members regarding the use of and dependency to nyaope are multi-faceted and can significantly impact various stakeholders and areas of interventions such as public policies, support services and societal attitudes. Understanding the perspectives of family members provides valuable insights into the challenges they face when dealing with a loved one's nyaope use and dependency.

### Implications for Community-Oriented Substance Use Programmes

For the development implementation and sustainability of programmes like COSUP, our findings carry significant relevance. Family-centred solutions, better family support, increased awareness and education, specialised treatment options, long-term recovery planning, collaboration between communities stakeholders, and cultural sensitivity are just a few of the key points. Effort should be specifically made to encourage users to seek rehabilitation from inside their own families rather than among their friends who are also users. These strategies can help prevent the formation of the nyaope brotherhood.

### Implications for healthcare, rehabilitation and social services: addressing perceived inadequacies

The findings of the study, notably the perception of shortcomings in social services, are likely to have ramifications for rehabilitation, and social service sectors. Addressing perceived gaps in social services is critical for providing individuals and families with comprehensive care and assistance. Collaboration among the healthcare, rehabilitation and social service sectors, as well as advocacy and enhanced training, can result in more effective and responsive services that address the requirements of those who are most in need. Our findings can be utilised to advocate for policy reforms and increased funding for treatment and support programmes for users. Policymakers can be kept up-to-date on the latest developments.

### Implications for the police and justice system: addressing perceived passivity.

Families afflicted by nyaope use and dependency have a bad perception of the police and justice systems, which has major implications for these institutions and nyaope treatment.

Improving communication between police and communities, law enforcement personnel and the judicial system, customised training for justice system professionals, exploring the establishment of specialised courts, and advocating for policy reforms within the justice system and collaboration with medical services should all be considered.

### Implications for schools

The fact that some parents were ignorant of their children's Nyaope use until the school informed them and that schools were reported as nyaope initiation platform sites has major consequences. Early detection, nyaope use education and preventative programmes, a secure school environment, and constant surveillance should all take place within this setting. Schools can help to lower the likelihood of nyaope use starting. Collaboration between school health services, the Department of Education and social services is necessary to reach that goal.

### Implications for further research

Further studies and confirmation of certain facets of users' experiences with nyaope will be guided by the outcomes of this study and the prior one (1). The focus in future will be on exploring and integrating the users' perspective into our qualitative understanding of nyaope addiction and then evaluating these factors further in a case control study.

### Conclusions

Families outlined the pathway for users of nyaope, from home to being disconnected to the family and eventually living on the streets as part of the 'nyaope brotherhood'. Many factors were identified that influence this pathway and also provide opportunities for prevention of nyaope addiction. The findings have implications for substance abuse programmes, social services, health and police services as well as schools.

### Author contributions

**Conceptualization:** Doudou Kunda Nzaumvila, Robert Mash, Toby Helliwell.

**Data curation:** Doudou Kunda Nzaumvila, Robert Mash, Toby Helliwell.

**Formal analysis:** Doudou Kunda Nzaumvila, Robert Mash, Toby Helliwell.

**Methodology:** Doudou Kunda Nzaumvila, Robert Mash, Toby Helliwell.

**Supervision:** Robert Mash, Toby Helliwell.

**Writing – original draft:** Doudou Kunda Nzaumvila.

**Writing – review & editing:** Robert Mash, Toby Helliwell.

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
