## [Decision Letter · Decision Letter 0]

31 Jul 2024

PONE-D-24-07000Family’s perceptions of their members who use Nyaope in Tshwane, South AfricaPLOS ONE

Dear Dr. Nzaumvila,

Thank you for submitting your manuscript to PLOS ONE. After careful consideration, we feel that it has merit but does not fully meet PLOS ONE’s publication criteria as it currently stands. Therefore, we invite you to submit a revised version of the manuscript that addresses the points raised during the review process.

We look forward to receiving your revised manuscript.

Kind regards,

Engelbert A. Nonterah, MD, PhD

Academic Editor

PLOS ONE

Reviewers' comments:

Reviewer's Responses to Questions

**Comments to the Author**

1. Is the manuscript technically sound, and do the data support the conclusions?

Reviewer #1: Yes

Reviewer #2: Yes

2. Has the statistical analysis been performed appropriately and rigorously? 

Reviewer #1: N/A

Reviewer #2: Yes

3. Have the authors made all data underlying the findings in their manuscript fully available?

Reviewer #1: No

Reviewer #2: No

4. Is the manuscript presented in an intelligible fashion and written in standard English?

Reviewer #1: Yes

Reviewer #2: Yes

5. Review Comments to the Author

Reviewer #1: The minor issues that need attention are the following:

Failure to avail data has been explained

line 28: it should be nyaope use.....

lines 60 to 64: the authors must also indicate that even the composition differs across samples because not all samples have milk, or paracetamol, etc. there are publications for such information

Nyaope is not a proper noun and should not be written in capital N, except at the beginning of a sentence

lines 130 to 131. it is not clear what linguistic discrimination refers to

line 452: the authors must explain what too late means

Reviewer #2: General comments

I very much enjoyed reading the manuscript. It is an interesting study of public health concern and deserves a scientific merit. The manuscript followed the structure of a journal article. The findings were arrived at through appropriate qualitative processes and analysis. However, the authors stated that “data cannot be shared publicly because of participants' privacy”. If necessary, the authors can still share the data with PLOS ONE in a more anonymized way. The authors can remove personal identifiers from the transcripts and that may help. If by any reason this is not possible, then it should be stated. For instance, participants during consenting may have been informed that the data (whether anonymized or not) would not be shared with any third party. If so, then it should be clearly stated as a reason for not sharing the data.

Below are some few minor comments for consideration.

The manuscript is well-written so I do not have much comments except for a few minor ones as below.

Data analysis

• The statement “The authors used the service of a professional linguist to translate the transcribed scripts from the vernacular languages into English and to transcribe the interviews verbatim” is not clear. Did the authors first have vernacular transcripts and then these transcripts translated into English? Or was the translation done while listening to the vernacular audios and transcribing straight ahead to English?

Ethical considerations

• Since the topic is sensitive, it would be interesting to demonstrate how Informed Consent was obtained. This could be in a brief sentence. For instance, having such an interview on such a topic could provoke emotions. This is a risk that should be explained to the participant when obtaining informed consent.

Results

• Sentence correction in line 203: Take off the first “are” in the sentence “…I would say that people who are using are Nyaope are dissimulators….”.

Strengths and limitations of the study

• This is a sensitive topic since it is about SUD. Couldn’t it have been better to use individual In-Depth Interviews rather than group interviews? Some persons might not bring out certain sensitive information in a group discussion. I think this could be a weakness/limitation of the study that is worth mentioning.

6. PLOS authors have the option to publish the peer review history of their article (what does this mean? ). If published, this will include your full peer review and any attached files.

**Do you want your identity to be public for this peer review?** For information about this choice, including consent withdrawal, please see our Privacy Policy .

Reviewer #1: No

Reviewer #2: **Yes: ** Aaron Kampim

---

## [Author Response · Author response to Decision Letter 1]

11 Oct 2024

The response to the reviewer is attached

---

## [Editor Report · Decision Letter 1]

2 Dec 2024

PONE-D-24-07000R1Family’s perceptions of their members who use nyaope in Tshwane, South AfricaPLOS ONE

Dear Dr. Nzaumvila,

Thank you for submitting your manuscript to PLOS ONE. After careful consideration, we feel that it has merit but does not fully meet PLOS ONE’s publication criteria as it currently stands. Therefore, we invite you to submit a revised version of the manuscript that addresses the points raised during the review process.

Amend your data availability statement in the submission system and on the manuscript Add the following headings prior to the references:1. Conflict of interest2. Funding information3. Author contribution4. Acknowledgements5. Data availability

We look forward to receiving your revised manuscript.

Kind regards,

Engelbert A. Nonterah, MD, PhD

Academic Editor

PLOS ONE
---

## [Editor Report · Decision Letter 2]

22 Jan 2025

Family’s perceptions of their members who use nyaope in Tshwane, South Africa

PONE-D-24-07000R2

Dear Dr. Doudou Kunda Nzaumvila,

We’re pleased to inform you that your manuscript has been judged scientifically suitable for publication and will be formally accepted for publication once it meets all outstanding technical requirements.

Kind regards,

Engelbert A. Nonterah, MD, PhD

Academic Editor

PLOS ONE
---

## [Editor Report · Acceptance letter]

PONE-D-24-07000R2

PLOS ONE

Dear Dr. Nzaumvila,

I'm pleased to inform you that your manuscript has been deemed suitable for publication in PLOS ONE. Congratulations! Your manuscript is now being handed over to our production team.

Kind regards,

on behalf of

Dr. Engelbert Adamwaba Nonterah

Academic Editor

PLOS ONE